# Intrauterine Thoracoamniotic Shunting of Fetal Hydrothorax with the Somatex Intrauterine Shunt: Intrauterine Course and Postnatal Outcome

**DOI:** 10.3390/jcm11092312

**Published:** 2022-04-21

**Authors:** Joleen Grandt, Ingo Gottschalk, Annegret Geipel, Ulrich Gembruch, Corinna Simonini, Eva Weber, Christoph Berg, Andreas Müller, Brigitte Strizek

**Affiliations:** 1Department of Obstetrics and Prenatal Medicine, University Hospital Bonn, Venusberg Campus 1, 53127 Bonn, Germany; joleeng@web.de (J.G.); annegret.geipel@ukbonn.de (A.G.); ulrich.gembruch@ukbonn.de (U.G.); corinna.simonini@ukbonn.de (C.S.); christoph.berg@ukbonn.de (C.B.); 2Division of Prenatal Medicine, Gynecological Ultrasound and Fetal Surgery, Department of Obstetrics and Gynecology, University of Cologne, 50923 Cologne, Germany; ingo.gottschalk@uk-koeln.de (I.G.); eva.weber@uk-koeln.de (E.W.); 3Department of Neonatology and Pediatric Intensive Care, Children’s University Hospital Bonn, 53127 Bonn, Germany; andreas.mueller@ukbonn.de

**Keywords:** fetal hydrothorax, fetal pleural effusion, fetal hydrops, thoracoamniotic shunting, fetal therapy

## Abstract

(1) Background: Severe fetal hydrothorax can be treated by intrauterine thoracoamniotic shunting (TAS). The aim of this study was to assess perinatal outcome and complication rates of TAS with a novel Somatex intrauterine shunt. (2) Methods: This is a single-center retrospective study of all fetuses with hydrothorax treated with TAS using a Somatex shunt between 2014 and 2020. (3) Results: A total of 39 fetuses were included in the study. Mean gestational age at first intervention was 27.4 weeks (range 19–33). Of these, 51% (*n* = 20) of fetuses had fetal hydrops, which resolved in 65% (13/20) before delivery. The live birth rate was 97% (*n* = 38), and 74% (*n* = 29) survived the neonatal period. The rate of postnatal pulmonary complications was high, with 88% of neonates requiring any kind of ventilatory support. There were 23% (*n* = 9) genetic abnormalities (trisomy 21 and Noonan syndrome). (4) Conclusions: TAS with a Somatex shunt has a high technical success rate, leading to high neonatal survival rates. Pregnancy and neonatal outcome is comparable to TAS for fetal hydrothorax using different shunt types.

## 1. Introduction

Fetal hydrothorax is the unspecific accumulation of fluid in the pleural cavity, which can occur both uni- and bilaterally [1]. One distinguishes between primary and secondary hydrothorax, depending on the etiology [2]. Primary hydrothorax has an incidence of approximately 1:15,000 pregnancies [3] and is defined as the accumulation of lymphatic fluid in the pleural cavity. Today, still, little is known about the causes that lead to the development of this disease [4]. Current research suggests that an abnormal development of the lymphatic system plays a role in the pathophysiology of primary hydrothorax [5]. Secondary hydrothorax is defined as the presence of a serous exudate in the pleural cavity due to infection, heart failure, aneuploidy or congenital malformations such as fetal diaphragmatic hernia or bronchopulmonary sequestration. Only in recent years, exome sequencing has revealed genetic variants that might play a role in the development fetal pleural effusion [6]. In both forms of hydrothorax, the accumulation of fluid in the pleural cavity leads to elevated intrathoracic pressure, which can lead to pulmonary hypoplasia and hydrops fetalis [1,4]. 

The clinical course of fetal hydrothorax ranges from spontaneous regression to a progressive course with the development of fetal hydrops, polyhydramnios and intrauterine death or a high risk of premature delivery [1,7]. Prenatal management options include repetitive thoracocentesis, thoracoamniotic shunting (TAS), pleurodesis with OK-432 [8] and premature delivery. The aim of TAS is to improve the circulatory situation of the fetus by decompressing the heart and lungs to counteract the development of hydrops and pulmonary hypoplasia [9]. Perinatal survival after TAS has been reported to be 58% for fetuses with and 73% without associated hydrops [10]. Accordingly, the regression of an existing hydrops after TAS is an important prognostic factor. Factors with a negative impact on fetal survival include the presence of polyhydramnios, mediastinal shift at initial examination, and a time interval of less than four weeks between initial shunt insertion and birth [10], but also smaller lung size after TAS [11]. Earlier gestational age at delivery is also considered to be a significant predictor of adverse outcome [12]. Factors that have been reported to have no prognostic value are the number of inserted shunts, hydrops fetalis at first presentation and gestational age at first shunt insertion [10,13], but these are not consistent in all of the studies.

Treatment of fetal hydrothorax by TAS appears to have a positive effect on perinatal survival, although several interventions might be required [10] as various complications can occur, the main one being dislocation of the shunt into the pleural or amniotic cavity. Other complications described in the literature are a blockage of the shunt [7], PPROM (preterm premature rupture of membranes) and preterm delivery [9]. Different types of shunts have been used in the past, but they have not been compared in terms of efficacy and safety profile. We therefore evaluated our experience in TAS with the Somatex intrauterine shunt that we have been using for TAS from 2014 and compared it to our own historic experience with the Harrison shunt.

## 2. Materials and Methods

### 2.1. Patients

To determine the effect of TAS using a Somatex Intrauterine shunt (Somatex Medical Technologies, Berlin, Germany), a retrospective analysis of all consecutive cases of primary fetal hydrothorax treated by TAS at the University of Bonn from 2014 to 2020 was performed. Exclusion criteria were secondary hydrothorax due to congenital pulmonary airway malformation (CPAM), congenital diaphragmatic hernia and lung sequestration. Patients were included even if they had TAS with a different type of shunt either before or following a Somatex shunt in our department. Reason for usage of a Harrison shunt was massive skin edema, where the Somatex shunt was deemed too short.

Diagnostic workup prior to TAS included a complete ultrasound assessment of the fetus, including transabdominal fetal echocardiography and Doppler sonography. Criteria for shunt intervention were either hydrops (defined as pleural effusion associated with skin edema, ascites or pericardial effusion), pleural effusion occupying more than 50% of the thoracic cavity for bilateral hydrothorax or mediastinal shift with complete deviation of the heart in the opposite half of the thorax in unilateral hydrothorax. Conventional karyotyping was recommended to all parents, and additional genetic testing was performed if clinically indicated. TAS was performed by specialists in fetal medicine with more than 10 years of experience in fetal shunting. All patients gave written informed consent for TAS after counseling, including the off-label use of the Somatex shunt for TAS.

Ethical approval was waived by the ethics committee of the University Hospital Bonn (No 546/20).

### 2.2. Outcome

The primary outcome parameter was perinatal survival, secondary outcome parameters were therapeutic effectiveness of TAS and procedural complications. To evaluate complication rates of the Somatex Intrauterine shunt, each individual shunt placement was assessed with regard to problems during shunt insertion, dislocation, occlusion/blockage or maternal complications.

### 2.3. Procedure and Study Protocol

A Somatex Intrauterine shunt was inserted in all patients under ultrasound guidance (Figure 1). 

The Somatex shunt is 25 mm long. The shunt has self-deploying parasols at both ends, it is preloaded into the 18-gauge insertion cannula. The inner diameter of the expanded shunt is 2.6 mm. The shunt consists of a nitinol wire mesh and internal impermeable silicone coating. In brief, the insertion needle is used to enter the fetal chest under ultrasound guidance, if possible perpendicular to the chest wall. The inner end of the shunt is then advanced by pushing the ejector forward until the parasol becomes visible and unfolds. The shunt is released by retracting the insertion cannula from the fetal chest; at this time, the second parasol unfolds in the amniotic cavity. Care must be taken not to retract the shunt with the insertion cannula. (See Appendix A).

Fetal anesthesia prior to shunting was used according to the treating physician’s choice depending on fetal position and movements. Tocolytics were only used when premature contractions occurred, and antibiotics were not given prophylactically. The procedure was generally performed without maternal anesthesia. In cases of bilateral hydrothorax, the placement of bilateral shunts was counted as one TAS procedure if it was performed during the same session but as two procedures if the second side was shunted on a different day. 

Follow-up by ultrasound was performed the next day and every 2–4 weeks to assess shunt position and evolution of hydrothorax and hydrops. In cases of dislocation or occlusion, a second shunt was placed if the above criteria for shunting still applied. Neonatal management regimen was not standardized between the various hospitals. 

Neonatal management included immediate clamping of the shunt and removal as soon as possible. The skin was closed with adhesive wound closure strips and compression bandage for approximately three days.

Results were compared to a previous study performed in our department that had evaluated the outcome of treatment of fetal hydrothorax using a Harrison shunt (Cook Medical Inc., Bloomington, IN, USA) [10].

### 2.4. Follow-Up

Maternal, fetal and neonatal characteristics were retrieved from the hospitals’ electronic database. For infants born in our hospital, postnatal records were reviewed. For infants born in other hospitals, discharge letters were reviewed after parental consent was obtained. Postnatal therapy included respiratory support, thoracic drain placement and medium-chain triglyceride-rich diet and, if necessary, total parenteral nutrition and octreotide/somatostatin in neonates with chylothorax. 

### 2.5. Statistical Analysis

Descriptive statistics were calculated using the Statistical Package for Social Sciences (SPSS 25.0, SPSS Inc., Chicago, IL, USA). Categorical variables were compared using the chi-square test or Fisher’s exact test as appropriate and student’s t-test for continuous variables. A Shapiro–Wilk test was used to test for normality of distribution. Numbers are presented as means if not stated otherwise. Significance was defined as *p* < 0.05.

## 3. Results

### 3.1. Patient Characteristics

From 2014–2020, a total of 56 fetuses underwent TAS with a Somatex shunt. However, 17 fetuses were excluded due to CPAM (*n* = 8), congenital diaphragmatic hernia (*n* = 6) or lung sequestration (*n* = 3). Thus, a total of 39 cases were included in the study. There were two twin pregnancies with only one affected fetus. Two patients were included after dislocation of a Harrison shunt. Mean maternal age was 33.7 years (range 21–47) and mean gestational age at diagnosis was 26.8 weeks (range 19–33 weeks). Out of all the fetuses in the study, 26 (66.7%) had bilateral hydrothorax and 33.3% (*n* = 13) had unilateral hydrothorax (6 right, 7 left). Meanwhile, 20 fetuses (51.3%) showed hydrops, 71.8% (*n* = 28) had polyhydramnios and 35.9% (*n* = 14) had a mediastinal shift at the time of diagnosis. (Table 1)

Conventional karyotyping was performed in 30 patients (76.9%). In seven fetuses, a genetic abnormality was found prenatally: there were four fetuses with trisomy 21 and three with Noonan syndrome. All parents opted for VAS after counselling if the genetic abnormality was known before the procedure, or to continue the pregnancy if the results became available after the procedure. Associated anomalies on ultrasound were muscular ventricular septal defect (*n* = 2), agenesis of ductus venous and partial agenesis of the cerebellar vermis (*n* = 1) and ventriculomegaly (*n* = 1).

#### 3.1.1. Details of TAS

Mean gestational age at first shunt insertion was 27.4 weeks (range 19–33). Amnion drainage was performed in eight (20.5%) patients. On average, 2.49 shunts were inserted per fetus during pregnancy (including Harrison shunts). Fetuses with a unilateral hydrothorax received 1.8 shunts on average (range 1–3), while fetuses with a bilateral hydrothorax received 2.8 shunts on average (range 1–6). Bilateral shunt placement was performed in 20 patients, in 15 patients at the same time and 5 in two separate procedures. More than one shunt procedure was necessary in 23 patients (58.9%), with 2 shunt procedures being performed in 19 fetuses and 3 shunt procedures in 4 fetuses. Reasons for repeat procedures were time delayed contralateral TAS in 5 in bilateral hydrothorax, shunt dislocation in 13 (including 3 dislocated Harrison shunts), and ineffective drainage and/or shunt occlusion in another 5 patients. Additional thoracocentesis was performed in 10 patients (25.6%) because of shunt occlusion or ineffective drainage despite correct position of the shunt. 

Hydrothorax resolved completely until birth in 38.5% (15/39) and hydrops resolved completely in 65% of the hydropic fetuses (13/20). Mean interval between first shunt insertion and delivery was 44.2 days (range 3–126).

#### 3.1.2. Complications of Somatex Shunts

Overall, 57 Somatex shunts were applied during the study period. In all patients, shunt placement was technically successful, but shunt placement was difficult in 19/57 procedures (33.3%). In one patient, the shunt remained stuck in the uterine wall. The further course of pregnancy was uneventful, and the shunt was removed at the time of cesarean section. In another patient, bleeding from a placental vessel during transplacental shunt insertion led to an emergency cesarean section. 

In the remaining patients, the shunt could not be positioned correctly initially, resulting in either (1) the inner end of the shunt not reaching the pleural cavity, (2) the outer end of the shunt ending within the skin of the fetus or (3) the shunt being deployed in the amniotic fluid due to fetal movement. In these cases, a second shunt was used to complete the procedure.

Dislocation after successful shunt placement occurred in 13 of 57 TAS procedures (22.8%) or 33.3% of patients, respectively. The mean interval from the insertion of a Somatex shunt to dislocation was 8 days (range 1–31). Dislocation occurred in 30.8% (4/13) into the amniotic cavity. In 69.2% (9/13), either the outer end of the shunt became embedded in the fetal subcutaneous layer, was overgrown by skin, or the entire shunt dislocated into the pleural cavity. In 17.9% (7/39), there was chorioamniotic separation, and one patient developed chorioamnionitis. Preterm premature rupture of membranes (PPROM) occurred in 15 (38.5%) pregnancies. 

### 3.2. Perinatal Outcome

There were 38 (97.4%) live births at a mean gestational age of 33.8 weeks (range 24–39 weeks). Three patients were lost to follow-up after being born alive.

Reason for delivery was PPROM or spontaneous onset of labor in 24 patients. Mean birth weight was 2530 g (±860 g; range 570–4370 g). Premature delivery before 37 weeks occurred in 76.3% (29/38) of the children, and almost half (*n* = 20) were born before 34 weeks. Thirteen children (34.2%) were born by vaginal delivery.

There was one intrauterine fetal death (IUFD) at 29 weeks GA, which occurred 12 days after bilateral shunting in a hydropic fetus, and there were nine neonatal deaths. All were born prematurely before 35 weeks, and the reason for neonatal death was lung hypoplasia and/or associated anomalies (Noonan syndrome, Leopard syndrome, trisomy 21) in eight and sepsis (after chorioamnionitis) in one neonate. Overall perinatal survival was 74.4% (29/39). Perinatal survival was higher (93.3%; 14/15) if hydrothorax resolved completely vs. 15/24 (62.5%) when it persisted, but this did not reach statistical significance (*p* = 0.057, OR 8.9, 95% CI 0.94 to 75.1). However, the survival of initially hydropic fetuses was considerably higher (12/13, 92.3%) if hydrops resolved before birth vs. 2/7 (28.6%) if it did not (*p* = 0.007, OR 30, 95% CI 2.19 to 411).

After birth, 31.4% of the neonates with known outcome (11/35) had respiratory distress syndrome, 88.6% (31/35) of newborns needed any kind of ventilatory assistance after delivery and 13 of these 31 developed pneumothorax, too. Eleven needed high-frequency oscillation ventilation, and one neonate required ECMO (extracorporeal membrane oxygenation). More than half (54.3%, 19/35) of fetuses had pulmonary hypertension after birth.

Pleural drain placement for residual pleural effusion was performed in 19 neonates. Surgical removal of a Somatex shunt was performed in 10 fetuses. Average length of hospital stay of the newborns was 27.6 days (range 8–63). 

Chylothorax could be confirmed as the underlying cause of hydrothorax in 38.5% (15/39). A genetic cause was confirmed in 9/39 (23.1%): four cases of trisomy 21, four cases of Noonan syndromes, and one case of Leopard syndrome. In 15 cases, the exact cause of hydrothorax remained undetermined after birth (idiopathic hydrothorax (*n* = 8), no residual pleural effusion after birth (*n* = 3), suspicion of a syndromic etiology due to associated central nervous system malformations (*n* = 2) and lost to follow up (*n* = 2). In one neonate, pulmonary stenosis was diagnosed after birth.

### 3.3. Comparison to a Historic Cohort Treated with Harrison Shunt

Compared to our historic cohort, there were no significant differences in patient characteristics in the Somatex vs. Harrison group for bilateral hydrothorax (66.7% vs. 61.5%), hydrops fetalis (51.3% vs. 35.9%) and trisomy 21 (12.9% vs. 18%). In addition, details of shunting did not differ significantly: the mean number of shunts inserted per fetus was 2.48 (range 1–6) vs. 2.53 (range 1–7) (Table 2).

The overall rate of shunt complications (dislocation and occlusion) was lower in the Somatex group in 46.2% (18/39) compared to 73% (57/78) in the Harrison group, which was statistically significant (*p* = 0.007), even if the individual rates of dislocation and occlusion did not differ significantly.

Complete regression of hydrothorax was significantly more frequent in the Somatex group (38.5%, 15/39) vs. 16.6% (13/78) in the Harrison group (*p* = 0.012). Resolution of hydrops did not differ significantly between the groups: 65% (13/20) in the Somatex vs. 71.4% (20/28) in the Harrison group (*p* = 0.76). There were also no significant differences for rates of preterm delivery (76% S vs. 66% H), live birth (97% S vs. 89% H) and overall survival with 74% (S) vs. 64% (H) respectively.

## 4. Discussion

Fetal hydrothorax is a heterogenous condition that can quickly progress to severe hydrops, which is then associated with a very high mortality of up to 53% [14].

In primary hydrothorax, spontaneous regression occurs in 22% of fetuses. Favoring factors are early diagnosis in the second trimester, unilateral hydrothorax as well as the absence of complications as hydrops or polyhydramnios [1,7]. Thoracoamniotic shunts were first used and described by Seeds and Bowes in 1986 [15], but data regarding efficacy and outcome are still inconsistent.

A recent systematic review of the effect of pleural-amniotic shunt insertion compared to conservative management in bilateral fetal hydrothorax without hydrops identified seven studies between 1992 and 2017 [16]. All studies were retrospective in nature, and only two of the seven studies allowed a direct comparison of conservative management versus shunting. The overall number of cases was small, and no difference in conservative management and shunt insertion could be demonstrated. A prospective study comparing these strategies is lacking.

In fetuses with hydrops, survival ranges from 12 to 24% in untreated cases [3,14]. If TAS is performed, the outcome is improved [12], but a study comparing different types of shunts does not exist. Our center had been using the Harrison shunt for TAS until the middle of 2014. We have previously published the outcome of 78 fetuses with fetal hydrothorax with (36%) and without (64%) hydrops after TAS with a Harrison shunt from 2002 to 2012 [10].

This is the first large cohort study evaluating TAS after the Somatex shunt became available in 2014. The technical success rate of TAS using a Somatex shunt and evolution during pregnancy are satisfactory: hydrops resolved in the majority (65%) of treated fetuses, and hydrothorax completely regressed until birth in 38%. Complications, however, occurred in a relevant number of patients: in 33%, there was dislocation and in 13%, the shunt was occluded. Chorionic membrane separation occurred in 17.9%, PPROM in 38.5%, and premature delivery in 76%. In more than half of the patients, more than one intervention was necessary: 48% of fetuses needed two and 10% needed even three interventions.

When comparing the current study to our own experience using a Harrison shunt [10], fetal characteristics were comparable between the groups, including no significant differences for bilateral hydrothorax and hydrops (Table 2). In addition, details of shunting did not differ significantly (Table 2). Overall, the rate of shunt complications (dislocation and occlusion) was lower in the Somatex group, but PPROM was more common (38.5% vs. 10.3%). Due to the smaller diameter of the shunt introducer, we expected to see a lower rate of PPROM. We can only speculate about the reason for the differences in PPROM rates; however, as there was no difference in terms of gestational age at birth or rate of preterm birth, we suspect that coding differences between the studies for reason of preterm birth might also play a role. 

Regression of hydrothorax until birth was observed significantly more often in the Somatex group (38.5%) vs. only 16.6% in the Harrison group, which was potentially due to the lower rate of shunt dislocation and occlusion. Overall, however, there was no difference in rates of live born fetuses and neonatal survival. 

In recent years, Witlox et al. published the outcome of 48 hydropic fetuses with hydrothorax treated with a double-pigtail silastic catheter between 2001 and 2016 [17], and Kelly et al. reported the outcome of 132 fetuses treated with TAS using a Rocket Stent between 1991 and 2014 [12]. In the study of Witlox et al., 85% of the fetuses were born alive at a median age of 34.4 weeks gestational age. Most (75%) of the fetuses were delivered preterm, which was associated with a higher mortality. Meanwhile, 60% of the fetuses had signs of pleural effusion at the time of birth, 70% needed ventilatory support, 53% needed mechanical ventilation and 75% of all fetuses survived the neonatal period. Survival rate of hydropic fetuses was 63% [17]. Kelly et al. reported that 87.8% of the fetuses with fetal hydrothorax were born alive at a median age of 35.4 gestational weeks. More than half (65%) of the fetuses were delivered preterm, which was associated with a higher mortality. Similarly, 65% of the fetuses needed ventilatory support, 53% needed mechanical ventilation and 65% of all fetuses survived the neonatal period. The survival rate of hydropic fetuses was 52% [12].

The results of our current study of TAS with a Somatex shunt are therefore comparable favorably to TAS with other shunt types: the live birth rate was 97%, and overall neonatal survival was 74%. Gestational age at delivery and rate of preterm birth were also similar to the literature (33.8 weeks and 76%, respectively). Neonatal pulmonary morbidity was equally high, with more than 88% of neonates requiring any kind of ventilatory support. The neonatal survival of hydropic fetuses was 70%.

Compared to the complication rates of other studies, the outcomes of TAS using a Somatex shunt seem to differ. Miyoshi et al. and Jeong et al. reported their experiences with a double-basket shunt (5 Fr, outer diameter 1.6 mm; Hakko Co., Nagano, Japan) with lower dislocation rates (5 and 6.6%) but higher occlusion rates (38 and 30.8%) [18,19].

In the largest published cohort (332 procedures from 1991–2020) after Rocket shunt insertion [20], shunt dislocation occurred in 9% of patients shunted for hydrothorax, or 5.4% of all inserted shunts. The place of dislocation was intrathoracic in 61% vs. 39% into the chest wall or amniotic fluid. Another study reported a re-intervention due to shunt dislodgement of a Rocket shunt in 1 of 27 patients (3.7%) [21].

These results differ from our own experience with both the Harrison and Somatex shunt. There is only one small study so far reporting the use of eight Somatex shunts in six patients, with similar shunt dislocation/occlusion rates of 50%; however, shunt placement was considerably earlier: around 22 weeks [22]. Nørgaard et al., reported their experience of 32 Somatex shunts (17 patients) in a letter [23], a re-intervention due to dislocation/occlusion was performed in 29% and complications during initial shunt placement were reported in four patients (23.5%). There was no PPROM within two weeks after the procedure and mean gestational age at delivery was reported to be 37 + 2 weeks in non-hydropic and 34 + 4 weeks in hydropic cases.

Considering all the results from the literature, we speculate that different shunt types have different and potentially specific complications; however, pregnancy and neonatal outcome was quite similar in all the studies, irrespective of the type of shunt used. Entrapment of the outer end of the Somatex shunt within the subcutaneous layer of the fetus seems to be a rather specific complication of this shunt type. We have seen this in fetuses with very thick skin edema or due to growth of the thorax during pregnancy.

Due to the small and retrospective design of our study, there are limitations. The small number of fetuses treated by very experienced physicians at a single center limits the generalizability of the results. On the other hand, it offers a unique opportunity to evaluate the outcome and complications of different shunt types, since mainly the same operators performed TAS during both study periods. However, even for those very experienced operators, correct placement of a Somatex shunt in the thorax seems to be more challenging compared to other shunt types, as approx. 30% of the procedures were difficult. To our knowledge, however, the rate of difficulties during shunt placement have rarely been reported in other studies. Interestingly, we did not observe this problem with the Somatex shunt for vesicoamniotic shunting (VAS). One difference is the gestational age at intervention, with VAS being performed earlier in pregnancy, mainly in the first or second trimester. Due to the advanced gestational age and polyhydramnios, fetal movements might impede TAS more. In addition, the length of the Somatex shunt is shorter (25 mm) compared to the Harrison shunt, which makes positioning fetuses with very thick skin edema more difficult (Figure 2). In view of the high complication rates of the Somatex shunt, technical improvements seems highly necessary. 

As we did not evaluate if there was a learning curve during the study period, we can only speculate that for inexperienced operators, TAS with a Somatex shunt might be even more challenging. Another limitation is that long-term data regarding fetuses treated with the Somatex shunt are still missing. In general, long-term outcome of intrauterine hydrothorax is considered favorably [12,24], although the rates of associated neurodevelopmental anomalies might be higher (up to 15%) than after other fetal interventions [25].

Fetuses with hydrothorax have an increased prevalence of genetic and other abnormalities. In accordance with the findings of previous studies, we found genetic abnormalities in 23% (9/39) of our patients (trisomy 21, Noonan syndrome and Leopard syndrome, a variant of Noonan syndrome). Careful and repeated ultrasound examinations are recommended, especially if hydrops persists despite drainage of the hydrothorax. These findings support the recommendation to perform not only karyotyping in all fetuses with hydrothorax but also a targeted search for Noonan syndrome and its variants and/or whole exome sequencing, especially in cases with severe hydrops.

Although the perinatal outcome of fetuses with hydrothorax has improved over the years, the difficulties of conducting prospective studies in fetal therapy as well as variations in shut approval and availability in different countries will prevent us from determining the optimal treatment approach for fetuses with hydrothorax in the near future. The current evidence allows physicians to choose the type of shunt they have the most experience with. At least in our center, the Somatex shunt seems to be slightly superior to the Harrison shunt for TAS, although complications rates, preterm delivery, and neonatal morbidity remain challenging.

## 5. Conclusions

TAS with a Somatex shunt is effective, leading to resolution of hydrops in 65% and complete regression of hydrothorax in 38.5%. In our study, 97% of treated fetuses were born alive, and neonatal survival was 74%. Difficulties during shunt placement, dislocation and occlusion led to re-interventions in more than half of the patients. The need for repeated shunt insertion, however, did not seem to be associated with adverse outcome.

Premature delivery and neonatal pulmonary morbidity remain problems after TAS, irrespective of the type of shunt used.

## Figures and Tables

**Figure 1 jcm-11-02312-f001:**
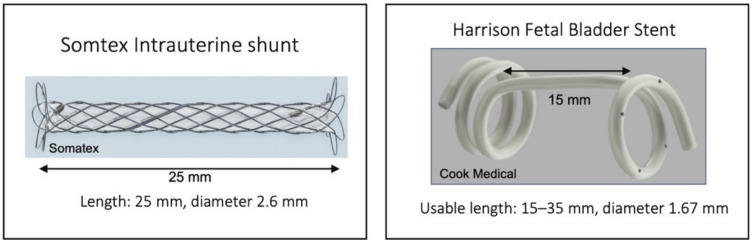
Shunt types used for thoracoamniotic shunting.

**Figure 2 jcm-11-02312-f002:**
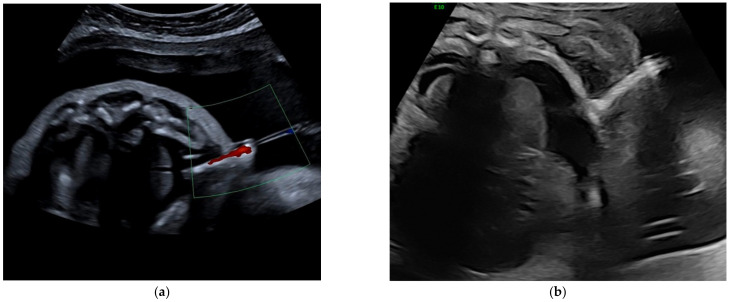
(**a**) Correct position of a Somatex shunt (axial view). Outward flow through the shunt is demonstrated by color Doppler. (**b**) Intracutaneous dislocation of a Somatex shunt in a fetus with massive skin edema on an axial view of the thorax. The inner end of the shunt is not reaching the pleural cavity.

**Table 1 jcm-11-02312-t001:** Patient characteristics of fetuses treated with thoracoamniotic shunting.

	*n* = 39
Maternal age	33.7 (21–47)
Gestational age at diagnosis (weeks)	26.8 (19–33)
Extent of pleural effusion	
unilateral	13 (33.3%)
bilateral	26 (66.7%)
Hydrops fetalis	20 (51.3%)
Polyhydramnios	28 (71.8%)
Major structural anomalies	2 (5.1%)
Prenatal genetic analysis	30 (76.9%)
Trisomy 21	4 (10.3%)
Noonan syndrome	4 (10.3%) *
Leopard syndrome	1 (2.6%) *
*Details of shunting*	
Gestational age at first shunt [weeks]	27.4 (19–33)
Unilateral shunt placement	19 (48.7%)
Bilateral shunt placement	20 (51.3%)
Mean number of shunts per fetus	2.49 (range 1–6)
Additional thoracocentesis	10 (25.6%)
Amniotic drainage	8 (20.5%)
*Intrauterine course and complications*	
Chorioamnionitis	1 (2.5%)
Chorionic membrane separation	7 (17.9%)
PPROM < 37 weeks	15 (38.5%)
Shunt dislocation	13/39 (33.3%)
Shunt occlusion	5/39 (12.8%)
Complete regression of hydrothorax	15 (38,5%)
Hydrops resolved	13/20 (65%)

Values are given in mean with range or *n* (%), PPROM preterm premature rupture of membranes. * Including postnatal diagnosis in one patient with Noonan syndrome and Leopard syndrome each.

**Table 2 jcm-11-02312-t002:** Comparison of patient characteristics and outcome after TAS according to type of shunt.

	Somatex *n* = 39	Harrison *n* = 78	*p*-Value
Mean gestational age at first shunt (weeks)	27.4 ± 3.2 (19–33)	26.5 ± 4.2 (16–33)	0.25
Hydrops	20 (51.3%)	28 (35.9%)	0.11
Bilateral hydrothorax	26 (66.7%)	48 (61.5%)	0.54
Polyhydramnios	28 (71.8%)	53 (67.9%)	0.83
Chorioamnionitis	1 (2.5%)	6 (7.7%)	0.12
Chorionic membrane separation	7 (17.9%)	6 (7.7%)	0.42
Complete regression of hydrothorax	15 (38.5%)	13 (16.6%)	0.012 *
Complete resolution of hydrops	13/20 (65%)	20/28 (71.4%)	0.76
Shunt dislocation	13/39 (33.3%)	36/78 (46.2%)	0.23
Shunt occlusion	5/39 (12.8%)	21/78 (26.9%)	0.1
Combined shunt related complication	18/39 (46.2%)	57 (73%)	0.007 *
PPROM	15 (38.5%)	8 (10.3%)	0.0009 *
IUFD	1 (2.6%)	9 (11.5%)	0.16
Gestational age at birth (weeks)	33.7 (24–39)	33.4 (23–40)	
Birth weight (g)	2530 ± 860	2358 ± 735	0.28
Live birth	38 (97.4%)	69 (88.5%)	0.16
Preterm delivery < 37 weeks	29/38 (76.3%)	46/69 (66.7%)	0.38
Neonatal death	9 (23%)	19 (24.4%)	0.87
Perinatal survival	29 (74.4%)	50 (64.1%)	0.26

* *p* < 0.05 was considered statistically significant; PPROM preterm premature rupture of membranes, IUFD intrauterine fetal death.

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
