# Peer review of "Intrauterine Thoracoamniotic Shunting of Fetal Hydrothorax with the Somatex Intrauterine Shunt: Intrauterine Course and Postnatal Outcome"

_jcm, 2022, doi:10.3390/jcm11092312_

Round 1

Reviewer 1 Report

this is an interesting and large series of TAS using a Somatex shunt. given the paucity of data using this shunt, the information reported is very valuable.

comments:

  1. for the average reader, a picture showing the various types of shunts (Somatex, Harrison, Rocket, pigtail etc) used for TAS would be very informative. I would strongly encourage the authors to add such a figure (with a centimeter next to the shunts to illustrate their length)
  2. could the authors speculate on the high rate of PPROM using the Somatex shunt?
  3. could the authors describe the management immediately after birth: do neonatologists need to clamp the shunt directly after birth or remove asap and then cover the wound from the drain with a specific plaster, etc etc?
  4. could the authors describe a few more results regarding the neonatal management including the % of use of Octreotide, % of using MCT diet, long-term neurodevelopmental outcome?
  5. clearly the Somatex seems to have various complications including occlusion, not being long enough in case of severe hydrops, or remaining stuck and needing to be removed surgically after birth, etc etc. I would think there is an urgent need for improvement here.

Reviewer 2 Report

The authors analyzed the perinatal outcome and complication rates of intrauterine thoracoamniotic shunting with a novel Somatex intrauterine shunt. They concluded that intrauterine thoracoamniotic shunting with a Somatex shunt has a high technical success rate leading to high neonatal survival rates.

Overall, well-designed and written study. However, I have several, mostly minor suggestions to improve the study:

1) Abstract – It is unusual for sentences to begin with a number (percentage). Similar is in the results section. E.g. ‘2.49 shunts were…’ Better formulation would be: ‘A total of 2.49 shunts were…’’ Please revise through the text.

2) For a clearer presentation please subdivide methodology in several sections: 2.1. Patients (including inclusion/exclusion criteria, Ethic Committee statements, indications for surgery …; 2.2.) Outcomes of the study and hypothesis; 2.3) Study protocol; 2.4.) Description of procedure; 2.5.) Follow-up; 2.6.) Statistical analysis

3) Description of the shunt is very briefly presented. Insertion of shunt should be described in more detail, in a way that one can repeat the process by reading this text. If available it would be interesting to add several pictures during the procedure.

4) Statistical analysis – Which statistical test was used to test normality of distribution? Please add.

5) Table 1 – The authors used the abbreviation ‘PPROM’ – Please provide explanation in the legend of the table.

Round 2

Reviewer 1 Report

i am satisfied with the revisions and wish to congratulate the authors for their efforts